# A Cross-Sectional Study: Structural and Related Functional Connectivity Changes in the Brain: Stigmata of Adverse Parenting in Patients with Major Depressive Disorder?

**DOI:** 10.3390/brainsci13040694

**Published:** 2023-04-21

**Authors:** Zhaowen Nie, Xinhui Xie, Lijun Kang, Wei Wang, Shuxian Xu, Mianmian Chen, Lihua Yao, Qian Gong, Enqi Zhou, Meng Li, Huiling Wang, Lihong Bu, Zhongchun Liu

**Affiliations:** 1Department of Psychiatry, Renmin Hospital of Wuhan University, Wuhan 430060, Chinalijunnkang@126.com (L.K.);; 2PET/CT/MRI and Molecular Imaging Center, Renmin Hospital of Wuhan University, Wuhan 430060, China; 3Taikang Center for Life and Medical Sciences, Wuhan University, Wuhan 430072, China

**Keywords:** major depression disorder, adverse childhood experiences, parenting, brain morphology, magnetic resonance imaging

## Abstract

**Background:** There is a high correlation between the risk of major depressive disorder (MDD) and adverse childhood experiences (ACEs) such as adverse parenting (AP). While there appears to be an association between ACEs and changes in brain structure and function, there have yet to be multimodal neuroimaging studies of associations between parenting style and brain developmental changes in MDD patients. To explore the effect of AP on brain structure and function. **Methods:** In this cross-sectional study, 125 MDD outpatients were included in the study and divided into the AP group and the optimal parenting (OP) group. Participants completed self-rating scales to assess depressive severity, symptoms, and their parents’ styles. They also completed magnetic resonance imaging within one week of filling out the instruments. The differences between groups of gender, educational level, and medications were analyzed using the chi-squared test and those of age, duration of illness, and scores on scales using the independent samples *t*-test. Differences in gray matter volume (GMV) and resting-state functional connectivity (RS-FC) were assessed between groups. **Results:** AP was associated with a significant increase in GMV in the right superior parietal lobule (SPL) and FC between the right SPL and the bilateral medial superior frontal cortex in MDD patients. **Limitations:** The cross-cultural characteristics of AP will result in the lack of generalizability of the findings. **Conclusions:** The results support the hypothesis that AP during childhood may imprint the brain and affect depressive symptoms in adulthood. Parents should pay attention to the parenting style and avoid a style that lacks warmth.

## 1. Introduction

Adverse childhood experiences (ACEs), intense stresses experienced by children, are implicated in subsequent psychological health [1,2,3]. Many children are exposed to ACEs, with about a quarter of individuals living in Europe and the United States having experienced at least one ACE [4]. The childhood experience is a critical determinant of health throughout life [5,6]. Studies have revealed that one ACE, adverse parenting (AP), is related to the risk of developing major depressive disorder (MDD) [7,8,9,10,11,12]. Gao et al. [13] explored the relationship between received parenting and the risk of MDD in 4567 Chinese women and found that lower levels of warmth and protectiveness and higher parental authoritarianism (mothers, fathers, or both) were significant risk factors for recurrent MDD. Long et al. [14] examined 1303 monozygotic twin pairs and found that parental authoritarianism may increase the risk of MDD, and Liu et al. [15] reported that lower levels of parental care were related to higher depression symptom scores.

However, the relationship between ACEs and depression is still debated. It has been hypothesized that ACEs can harm children both directly (e.g., physical abuse, emotional neglect) and indirectly (e.g., parental conflict and AP [6,16]), creating a stressful environment for development [6]. It has been postulated that these events remodel brain development and might leave traits detectable by neuroimaging [17]. This hypothesis seems plausible since childhood is a vulnerable period of neural development [18,19,20], and some data already support this. Childhood adversity was associated with amygdala and hippocampus volume differences [21], while Miguel et al. [20] detected an association between an early adverse environment and changes in brain volume and connectivity, particularly in the amygdala and hippocampus. Childhood maltreatment appears to have significant negative associations with right amygdala and anterior hippocampal volumes [22].

In addition to affecting brain structure, ACEs might also affect brain function and resting-state functional connectivity (RS-FC). RS-FC is an index that measures the degree of temporally synchrony of the blood oxygenation level-dependent signal among different brain regions. Yeo et al. organized a 7-network parcellation based on distinct characteristics of different cortical regions and data from 1000 subjects [23]. Despite the RS-FC raising some initial skepticism, it has withstood extensive experimental work over the years. RS-FC can not only explain some behavioral outputs, such as sustained attention, severity of psychiatric symptoms, and personality traits, but can also differentiate subjects with accuracy [24,25,26]. For example, childhood neglect is related to weaker amygdala RS-FC, with clusters detected within the left dorsal precuneus [27] and increased RS-FC between the salience network (SN) and the default mode network (DMN) [28]. Early-life stress is positively associated with the within-network connectivity of the ventral attention network (VAN), the dorsal attention network (DAN), and the between-network connectivity of the VAN-DAN [18]. Both ACEs and internalizing symptoms were related to a decreased anticorrelation between the DMN and the DAN [29], and positive parenting was associated with decreased coactivation of the superior parietal lobule (SPL), a component of the DAN, with the executive control network [29]. The DMN plays a vital role in cognitive and social functions, such as self-referential processes, and the FC within the DMN is associated with positive parenting traits [30,31]. The DAN and the VAN are specialized for disparate subprocesses of attention but tend to cooperate to maintain normal cognitive processes [32]. The RS-FC within and between the VAN and the DAN is related to early-life stress [18].

Of all the ACEs, parent–child bonding styles that make the child feel unloved when growing up may not be regarded as serious as other ACEs. Nevertheless, they are common in the population exposed to ACEs [33]. Despite their prevalence, there is a serious underrepresentation of studies on AP, perhaps because it is viewed as a “lesser” ACE. Only a few small, heterogeneous studies have explored the association between AP and brain alterations, and AP (such as overprotection) may be negatively correlated with the gray matter volume (GMV) of the hippocampus [34] and the dorsolateral prefrontal cortex (DLPFC) [35]. Similarly, while patients with MDD are commonly affected by ACEs, there have been no AP-relevant neuroimaging studies in this population, and there is a need for robust data on the impact of AP on the development of depression.

Here we examined the association between AP and multimodal neuroimaging features in patients with MDD participating in the Early Warning System and Comprehensive Intervention for Depression (ESCID) project [36,37]. We hypothesized that childhood AP was related to a decrease in GMV of the amygdala and hippocampus and changes inter- or intra-FC of some specific brain networks, such as the DMN, the DAN, and the VAN. Our major objectives were to explore possible structural and inter- or intra-network FC changes occurring in MDD patients exposed to AP. First, we measured the GMV within the whole brain and then compared it between the optimal parenting (OP) group and the AP group. We hypothesized that the differential brain region of GMV was the amygdala or hippocampus. Then we calculated the inter- and intra-FC of the DMN, the DAN, and the VAN. Our exploratory objectives were to examine relationships between neuroimaging changes and clinical and cognitive features. We estimated the relationship of the parental bonding instrument (PBI) scores with the values of regional GMV and FC to investigate the effect of parenting style on brain development.

## 2. Methods

### 2.1. Study Design, Participants, and Inclusion and Exclusion Criteria

This observational study was undertaken at the Renmin Hospital of Wuhan University, following the principles of the Declaration of Helsinki [38]. The Ethics Committee of Renmin Hospital of Wuhan University, Wuhan, Hubei, China, approved the study protocol. Written informed consent was obtained from all participants. This report was written following the STROBE statement [39].

MDD patients were selected from the Early Warning System and Comprehensive Intervention for Depression (ESCID) project. We calculated the sample size using G*Power software (latest ver. 3.1.9.7; Heinrich-Heine-Universität Düsseldorf, Düsseldorf, Germany). Informed consents for 8 participants were withdrawn, and 11 participants did not meet the inclusion criteria. One hundred and twenty-five MDD outpatients were recruited between May 2019 and April 2022. The inclusion criteria were: (1) being aged 18–50 years; (2) meeting DSM-5 diagnostic criteria for MDD diagnosed by two experienced psychiatrists; and (3) having a junior high school education or higher. Exclusion criteria were: (1) mental health disorders other than MDD diagnosed according to the DSM-5; (2) a history of organic brain disease; (3) severe stupor or other symptoms that could interfere with the study; (4) transcranial magnetic stimulation (TMS) or electroconvulsive therapy (ECT) treatment within the prior six months; and (5) pregnancy (Figure 1).

### 2.2. Assessment Instruments and Grouping Criteria

Enrolled patients completed the following instruments: the PBI [40], the Hamilton rating scale for depression (HAMD-17) [41], the patient health questionnaire-9 (PHQ-9) [42], the Hamilton Anxiety Scale (HAMA) [43], the Snaith–Hamilton pleasure scale (SHAPS) [44], and the digit symbol substitution test (DSST) [45] on the ESCID website. Magnetic resonance imaging was then completed within one week of completing these instruments.

The grouping criteria for optimal parenting (OP) and AP were modified from the studies of Parker and Margaret [46,47], who regarded high care and low overprotection as OP and affectionless control (low care, high overprotection) as AP. Some studies have suggested that there are cultural differences in parenting between western countries and China [9,13]. Contrary to western countries, parental protectiveness is positively related to warmth and decreases the risk of MDD in China. The effect of protectiveness is considered a representation of parental warmth and lovingness rather than increasing the risk for major depression as it does in the West. Consequently, we chose the relatively representative items of warmth and authoritarianism as grouping items, so the AP group was defined as having low warmth and high authoritarianism and the OP group as having high warmth and low authoritarianism according to the mean parental warmth and authoritarianism scores.

The PBI was used as a measure of participants’ attitudes toward their parents’ parenting style and to generate the AP and OP groups (with OP as the control group). The PBI, which contains 16 items, was modified by Kendler [40] and assesses warmth, protectiveness, and authoritarianism in both mothers and fathers before their children are 16 years old. PBI scores are not influenced by depressive states [48]. In total, data from 125 patients who completed four PBI item scores, including parental warmth and authoritarianism, were subjected to hierarchical cluster analysis by the Ward method to generate two clusters. In addition to the PBI, the clinician-rated HAMD-17 and self-rated PHQ-9 were used to evaluate depressive symptoms, SHAPS for anhedonia, and DSST for cognitive function.

### 2.3. MRI Acquisition

MRI data were acquired at the PET Center of Renmin Hospital of Wuhan University using a 3T scanner (General Electric, Milwaukee, Brookfield, WI, USA). T1-weighted structural images were acquired with the following parameters: repetition time (TR) = 8.5 ms; echo time (TE) = 3.2 ms; inversion time = 450 ms; flip angle (FA) = 12°; visual field (FOV) = 256 mm × 256 mm; matrix size = 256 × 256; slice thickness = 1 mm; slice gap = 0 mm; and locs per slab = 180. Subjects were kept in a comfortable position to keep them quiet, calm, and moving as little as possible with closed eyes. Resting-state BOLD fMRI data was acquired by employing spin echoplanar imaging (EPI) sequences with the following parameters: 212 times; 32 slices; slice thickness = 3.0 mm; slice gap = 0 mm; interval = 1 mm; TR = 2000 ms; TE = 30 ms; FA = 90°; matrix size = 64 × 64; and FOV = 240 mm × 240 mm.

### 2.4. MRI Data Processing

The VBM8 toolbox (http://dbm.neuro.uni-jena.de/vbm8/, accessed on 2 April 2022) in Statistical Parametric Mapping 8 (SPM 8; https://www.fil.ion.ucl.ac.uk/spm/software/, accessed on 2 April 2022) was used to preprocess the structural imaging data. Preprocessing included data conversion, quality testing, segmentation and normalization, index extraction, quality re-testing, and smoothing. DICOM images were converted to the NIFTI format for processing. The brain was separated into gray matter, white matter, and cerebrospinal fluid after segmentation. Images were normalized with the diffeomorphic anatomical registration through the exponentiated lie (DARTEL) algorithm to the Montreal Neurological Institute (MNI) template. We then extracted the GMV of all subjects. All structural images were smoothed with an isotropic Gaussian kernel of 8 mm at full width at half maximum (FWHM).

The resting-state fMRI (rs-fMRI) data was preprocessed using the RESTplus v1.2 toolbox in SPM 12. After data conversion, the first 10 volumes were deleted to ensure steady state. In addition, slice timing was carried out for time-level correction. Spatial-level correction consisted of realignment and normalization. We excluded subjects with excessive head movement (>3 mm or >3°) according to the realignment parameters. Six participants were excluded from each of the AP and OP groups, and 113 participants were included in the subsequent analysis. All functional images were smoothed with a 6 mm FWHM Gaussian filter. Then, we performed detrending, nuisance covariate regression, and filtering (0.01–0.08 Hz). All operations were conducted on the MATLAB R2013b platform (MathWorks, Sherborn, MA, USA).

### 2.5. Voxel-Wise FC Analyses

Wake Forest University’s (WFU) PickAtlas 3.0.5b [49] was used to obtain region of interest (ROI) templates in the MNI space. FC analysis was completed using RESTplus v1.2 after data preprocessing. Brain regions in which GMV differed significantly between the AP group and OP group were selected as seed points (ROI 1) in the voxel-wise FC analysis. To increase the robustness of the FC results, we performed a sensitivity analysis as follows: a sphere with a radius of 6 mm centered on the coordinates of ROI 1 was selected as the region of interest (ROI 2) using WFU PickAtlas 3.0.5b. The average time series of all the voxels in ROIs 1 and 2 was extracted and calculated. Then, voxel-wise Pearson’s correlation coefficients were calculated on the time series between ROI 1/2 and DMN [23] and then converted to *Z* values by Fisher’s equation to generate subject-specific maps [50]. Thus, the *Z* value representation of the functional connections between ROIs and the DMN were obtained. We used the same approach to analyze the intra-network FCs.

### 2.6. Statistical Analyses

Differences in sex and education levels between the AP and OP groups were assessed by chi-squared analysis. The independent samples *t*-test was applied to explore between-group differences in age, duration of illness, and PBI, PHQ-9, and HAMD-17 scores. Pearson’s correlation coefficients were used to determine correlations between GMVs and PBI item scores. A *p*-value of <0.05 (two-tailed) was deemed statistically significant. All analyses were conducted in IBM SPSS Statistics v25.0 (IBM Statistics, Armonk, NY, USA). GMV and FC analyses were executed in SPM 8 and SPM 12 separately using the general linear model. To control for confounding effects of age, gender, level of education, duration of illness, and head motion parameter, these factors were included in the calculation as covariates. The Gaussian random field (GRF) correction (voxel *p*-value < 0.001, cluster *p*-value < 0.05, one-tailed) was performed to correct for multiple comparisons using the RESTplus v1.2 toolbox [51,52].

## 3. Results

### 3.1. Clinical Characteristics

After hierarchical clustering, 85 patients were assigned to the AP group and 40 to the OP group. As expected, there were statistically significant differences in the four PBI items between the two groups (*p* < 0.05). There were no significant differences in age, gender, education level, medications, HAMD-17, PHQ-9, HAMA, SHAPS, DSST, or duration of illness between the two groups (Table 1).

### 3.2. Differences in GMV

Compared with the OP group, the AP group showed increased GMV in a cluster that had maximally intense voxels at MNI coordinates *x* = 30, *y* = −67.5, *z* = 55.5 (74 voxels, *T* = 3.892, *p* < 0.001, GRF correction) in the right SPL (Figure 2).

### 3.3. Differences in Functional Connectivity

The differential brain region of the GMV analysis between two groups is the right SPL. Although the SPL is not the predicted region of structural change, it is a part of DAN. We hypothesized that childhood AP is related to changes in the inter-FC of the DAN and the DMN. Consequently, we calculated the FC between the SPL and the DMN. In the DMN, voxel-wise FC between the right SPL and the right smFC (Cluster 1), the right smFC (Cluster 2), and the left smFC (Cluster 3) were significantly different between the AP group and the OP group (Table 2). Maximally intense voxels were at MNI coordinates *x* = 6, *y* = 57, *z* = 3 (13 voxels, *T* = 4.762, *p* < 0.001 GRF correction) for the first cluster (Figure 3a), the second cluster (Figure 3b) had maximally intense voxels at MNI coordinates *x* = 0, *y* = 60, *z* = 15 (13 voxels, *T* = 4.092, *p* < 0.001 GRF correction), while the third cluster (Figure 3c) had maximally intense voxels at MNI coordinates *x* = −9, *y* = 51, *z* = 45 (19 voxels, *T* = 4.069, *p* < 0.001 GRF correction). Using a second analytical approach to set the ROI and test robustness, the results were similar (see Appendix A).

As our hypothesis mentioned, AP may relate to changes in the inter- or intra-FC of some specific brain networks, such as the DMN and the DAN. To further investigate the effect of parenting style on the FC of the DAN and the DMN, we analyzed intra-network connectivity within the DMN, the VAN, and the DAN. No cluster survived the intra-DAN and intra-VAN FCs. Compared with the OP group, the intra-DMN FC showed an increase in a cluster with maximally intense voxels at MNI coordinates *x* = 48, *y* = −42, *z* = 3 (43 voxels, *T* = 5.509, *p* < 0.001, GRF correction) in the right middle temporal gyrus (MTG) (Figure 4).

### 3.4. Exploratory Data Analyses

As shown in Figure 5, the GMV of the ROI was significantly negatively correlated with maternal warmth scores (*r* = −0.329, *p* < 0.001) and positively correlated with maternal authoritarianism scores (*r* = 0.196, *p* = 0.037) and paternal authoritarianism scores (*r* = 0.195, *p* = 0.038).

To further explore the relationship between changes in FC and specific parenting styles, we performed a Pearson correlation. We found the FC signal between the right SPL region and the smFC was negatively correlated with parental warmth scores (*r* = −0.368~−0.207, *p* < 0.05). The FC signal between the left smFC and the right MTG is negatively correlated with maternal warmth scores (*r* = −0.504, *p* < 0.001) and paternal warmth scores (*r* = −0.219, *p* < 0.05), while positively correlated with maternal authoritarianism scores (*r* = 0.251, *p* < 0.001). (See Appendix A).

We additionally calculated correlations between PBI item scores and depressive symptoms such as anhedonia, sleep disturbances, psychomotor retardation, and anxiety/somatization. There was a positive correlation between paternal authoritarianism and anxiety/somatization scores but no significant results in the other symptoms (see Appendix A). Although AP and brain structure were significantly correlated, the interactions between AP, GMV, and depressive symptoms are also worth exploring in future analyses.

## 4. Discussion

Here, we report differential brain structure features as detected by rs-fMRI mainly in the right SPL and associated functional alternations in the bilateral medial superior frontal cortex (smFC) and right MTG in MDD patients experiencing AP or OP. These imaging traits were further associated with AP characteristics.

### 4.1. AP in MDD

There are profound cross-cultural differences in parenting styles [53] and a notable difference between Eastern and Western cultures [54]. The values and characteristics of parenting are specific to Eastern cultures, in particular differences in how cultures evaluate parental care. The most representative is overprotectiveness. High overprotectiveness is believed to be a component of affectionless control in western populations, where people attach great importance to individuality and privacy [55,56,57]. In Eastern cultures, people tend to foster collectivism, regard the family as a single unit, and think highly of the maintenance of emotional bonds between family members [58,59]. Therefore, Eastern cultures consider parental over-interference or protection as caring and protective [13,58,59]. Given the controversy over high overprotectiveness and the cross-cultural consistency of warmth, we treated warmth rather than high overprotectiveness as a feature of OP. Furthermore, to acquire a relatively reliable and culturally specific grouping, we defined the AP and OP groups using hierarchical cluster analysis, as previously described [54,58,59]. The results of our grouping are similar to those of Hiroko et al. [60], with 32% of MDD patients evaluated classified as OP in our sample.

### 4.2. Increased GMV of the Right SPL, the Stigmata of Childhood AP?

We found that markedly increased GMV in the right SPL was associated with AP. Previous studies have suggested that AP experiences during childhood and adolescence generate chronic hypothalamus-pituitary-adrenal (HPA) axis hyperactivity, resulting in abnormal neurodevelopment in the normal (non-depressed) population; for example, reduced GMV in the hippocampus [61,62]. Exposure to AP could activate the release of corticotropin-releasing hormone (CRH) and induce increased production of glucocorticoids from the HPA axis and higher levels of proinflammatory cytokines in the brain [63,64]. The glucocorticoids and proinflammatory cytokines are related to brain abnormalities in depression [65]. A study of 50 healthy young adults suggested that lower parental care and higher parental overprotection scores, namely AP, correlated with reductions in the GMV of the DLPFC [35]. Interestingly, our study in MDD patients showed a positive association between the GMV of the right SPL and AP. This finding allows us to formulate another valuable hypothesis for comprehending how early adversity modifies brain structure. AP may affect the normal procedure of synaptic pruning during childhood, resulting in increased GMV of the right SPL. Pruning is a part of brain maturation, which occurs during a sensitive period in late childhood and early adolescence [66,67,68,69,70]. The role of pruning is to strengthen the most frequently used connections and remove infrequently used connections to make the brain circuits more efficient [71]. Pruning is a critical feature of brain maturation, benefiting reasoning processes [72]. There are plenty of reports that synaptic pruning induces decreases in parietal GMV and density before adulthood [73,74,75,76]. The thinning of the superior parietal cortex is associated with age-related improvements on a working memory task, thought to result from selective pruning of inefficient synaptic connections [77,78]. It has previously been observed that thinner parietal cortices were associated with improved verbal learning and memory, visuospatial functioning, and spatial planning and problem solving [79]. Sowell et al. consistently found that longitudinal cortical thickness development was negatively related to vocabulary knowledge in the left dorsal frontal and parietal lobes [80].

Early-life adversity is known to alter the trajectories of structural and functional brain maturation by impairing synaptic pruning [72,81,82]. Abnormalities in neuroendocrine and neuroinflammation cellular and molecular mechanisms caused by early-life adversity may be related to disruption of pruning [83,84,85]. Our finding may reflect that we conducted the study in patients with MDD, and indeed, our result is similar to that reported by Akemi et al., who performed their study in patients with higher depressive levels and found that exposure to parental verbal abuse was associated with increased GMV in the superior temporal gyrus [86]. Hence, exposure to AP may interfere with neurodevelopment by impairing pruning. This is consistent with our findings: as expected, there was a significant negative correlation between the ROI GMV and the mother’s warmth scores and a positive correlation between both parents’ authoritarianism scores. We therefore presume that the worse the parenting style, the larger the GMV. However, the potential mechanism of AP affecting brain development requires more and further study to illuminate.

### 4.3. Differences in Functional Connectivity

Brain structure and function are closely related, and structure serves as the foundation for function. In addition to the effects on brain structure, there were also changes in the FCs between the right SPL and bilateral smFC in patients exposed to AP. The SPL, which is part of the attention and executive control networks, coordinates attention under competing conditions and voluntary attention [87] and also plays a key role in attention shifting [88,89,90]. The smFC, including the supplementary motor area (SMA) and presupplementary motor area (preSMA), is related to movement and cognitive control [91]. Increased neural activity in the SMA and precuneus is closely associated with the execution of attentional shift between object features, and these regions may play a critical role in the cognitive process of nonspatial attentional shift [92]. Attention-shifting dysfunction is also a feature of people exposed to AP. Elena et al. showed that greater maternal negative behavior could be associated with a decreased ability to disengage attention from negative stimuli [93]. Parental depressive symptoms, a more negative family atmosphere, and maternal authoritarianism predicted children’s inattentive symptoms [93]. The dysfunctional FC between the SPL and smFC also prompts us to consider attention shifting. Attention, as a complex cognitive function, is closely related to the DAN [94] and DMN [95], especially the FC between the DAN and DMN [96,97,98]. Two attentional states have been proposed: one that relies on the DMN and is relatively effortless and ideal for sustaining attention (such as when resting or watching a movie), and another that relies on the DAN, which is effortful and more suited to selective or executive attention (such as when performing a task) [99]. Christiane and colleagues reported that a stronger FC between the DAN and the DMN was associated with weaker attention switching traits [100]. Similarly, we found that childhood AP was related to an increased FC between the right SPL (part of the DAN) and bilateral smFC (part of the DMN). Alterations in DMN activity have been consistently reported in MDD [101,102]. The intra-DMN FC increased in the AP group. The relationship between ACEs and the intra-DMN FC is consistent with previous findings. For example, Keila et al. [103] emphasized that the family environment may be associated with differences in intrinsic DMN FC, with an unharmonious family environment associated with infants exhibiting stronger connectivity between two core DMN regions [104]. The DMN may be a crucial marker of the effects of ACEs and early-life stress closely related to emotional development and subsequent psychological health.

### 4.4. Limitations

Our study has some limitations. First, the cross-cultural characteristics of AP [13] will affect the generalizability of the findings, and we acknowledge the difficulties in comparing Eastern and Western countries. Hence, a suitable instrument to enhance comparability and promote cross-cultural polycentric research is needed to address this limitation. Second, we failed to find a statistically significant difference in DLPFC [35] and hippocampus [34] characteristics previously reported for general populations exposed to AP. This may be because MDD patients have pre-existing changes that mask the effects of AP or because our sample size was insufficient to detect differences in these brain regions. Third, due to the cross-sectional nature of the study and limitations of attention testing, we cannot confirm a causal relationship between AP, attention shifting, and our findings. Nevertheless, our study is hypothesis-generating, namely, that specific alterations in brain structure and function may be stigmata of AP in patients with MDD. Fourth, as a retrospective study, we did not further explore possible peripheral biomarkers associated with changes in brain imaging. For instance, Kosuke et al. [34] detected a significant association between parental overprotection scores, cortisol responses, and hippocampal GMV. Fifth, it is unethical not to give timely medication to patients with MDD, and there may be selection bias if only unmedicated patients are included in cohort studies. For these reasons, we did not apply an exclusion criterion with respect to the patients’ medications. Instead, we analyzed medication status between the AP and OP groups, and no significant differences were found.

## 5. Conclusions

Here we found that AP, as a low warmth and high authoritarianism parenting style during childhood, is related to significant enlargements in the right SPL and increased FC between the right SPL and the bilateral smFC in MDD patients. A more well-designed longitudinal study with a larger sample from different cultural backgrounds is essential to verify the effect of AP. The findings further emphasize the importance of optimal and appropriate parenting and arouse the attention of society and families to the parenting style. Our study generates an intriguing hypothesis that brain changes may be the imprint of the childhood AP that affects certain depressive symptoms.

## Figures and Tables

**Figure 1 brainsci-13-00694-f001:**
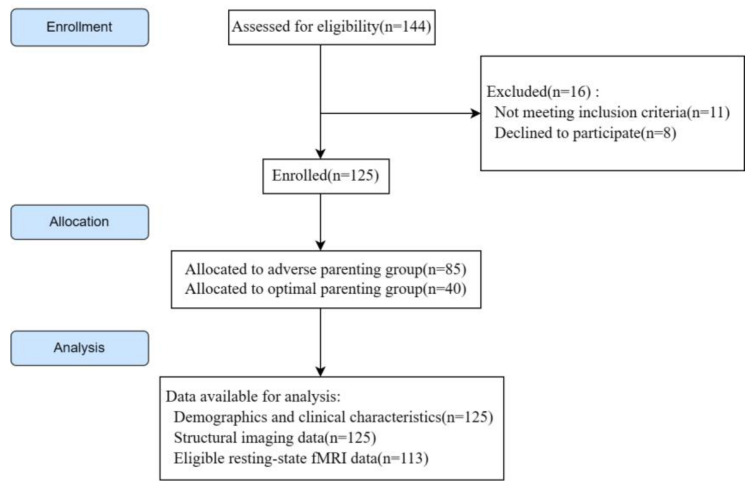
Flowchart representing the procedure of study design and participants.

**Figure 2 brainsci-13-00694-f002:**
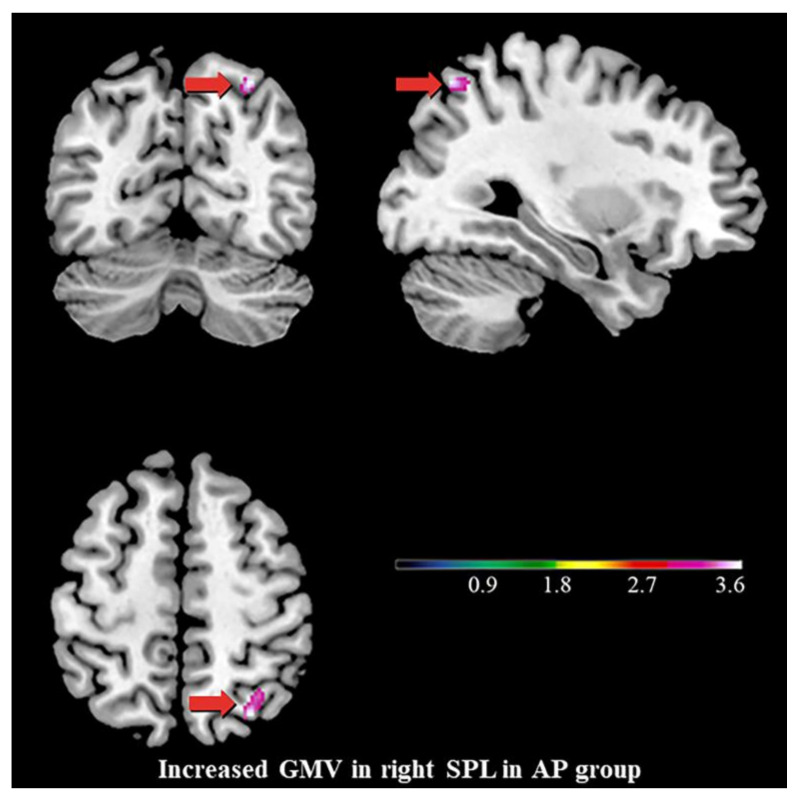
Significant difference in GMV between the AP and OP groups. This cluster contained 74 voxels, mainly in the right SPL, with a peak *T* value of 3.892. Significant at voxel *p* < 0.001, cluster *p* < 0.05, corrected by the GRF. The red arrow shows the corresponding brain region.

**Figure 3 brainsci-13-00694-f003:**
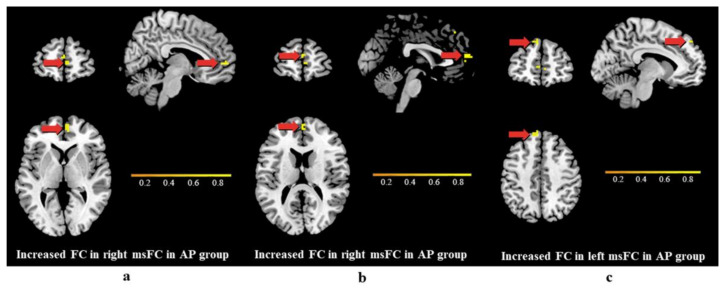
The images show increased FC from the right SPL (ROI 1) to the bilateral medial superior frontal cortex (smFC) in the AP group compared with the OP group at rest. In the DMN, voxel-wised FC between the right SPL and the right smFC were significantly different between the AP group and the OP group (**a**). In the DMN, voxel-wised FC between the right SPL and the right smFC increased in the AP group (**b**). In the DMN, voxel-wised FC between the right SPL and the left smFC also increased in the AP group (**c**). The red arrow shows the corresponding brain region. The right SPL was set as the seed because the GMV of this region was increased in the AP group (see Figure 2).

**Figure 4 brainsci-13-00694-f004:**
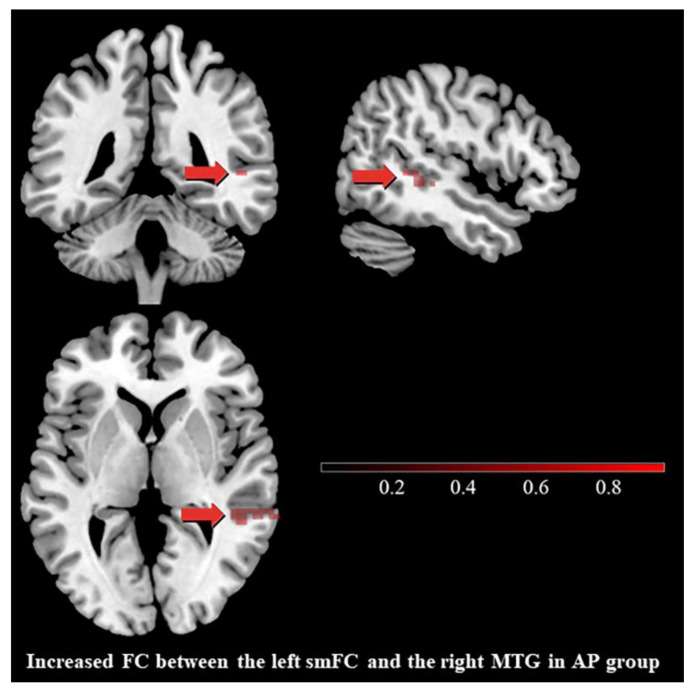
In the DMN, differences in FC between groups between the left smFC and the right MTG. The red arrow shows the corresponding brain region. The positive value (red) means OP < AP at a threshold of *p* < 0.001, GRF-corrected.

**Figure 5 brainsci-13-00694-f005:**
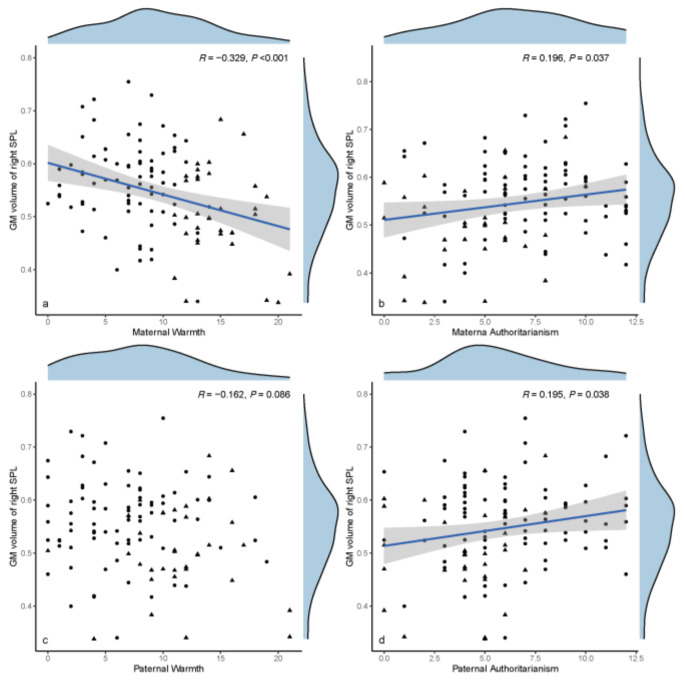
A negative correlation between the GMV of the ROI and maternal warmth scores (*r* = −0.329, *p* < 0.001) and a positive correlation between the GMV of the ROI and parental authoritarianism scores (for mother: *r* = 0.196, *p* = 0.037; for father: *r* = 0.195, *p* = 0.038) were observed. The GMV of the ROI is negatively correlated to maternal warmth scores (*r* = −0.329, *p* < 0.001) (**a**). The GMV of the ROI is positively correlated to maternal authoritarianism scores (*r* = 0.196, *p* = 0.037) (**b**). There is no significant correlation between the GMV of the ROI and paternal warmth (**c**). The GMV of the ROI is positively correlated to paternal authoritarianism scores (*r* = 0.195, *p* = 0.038) (**d**). The triangle represents the AP group and the circle represents the OP group. The density curves represent the distribution of the variables.

**Table 1 brainsci-13-00694-t001:** Demographics and clinical characteristics.

Characteristics (Mean ± SD)		Adverse Parenting	Optimal Parenting	*χ* ^2^ */t*	*p*
		*n* (%)	*n* (%)		
Gender	Female	67 (78.8%)	26 (65.0%)	2.729 *^a^*	0.099 *^a^*
	Male	18 (21.2%)	14 (35.0%)		
Education level	High school or	6 (7.1%)	4 (10%)	0.339 *^a^*	0.844 *^a^*
	less				
	Undergraduate	65 (76.4%)	30 (75%)		
	Postgraduate or	14 (16.5%)	6 (15%)		
	higher				
Medications	Drug-naïve	50(58.8%)	23(57.5%)	0.020 *^a^*	0.889 *^a^*
Age (years)		25.6 ± 6.2	24.7 ± 5.4	0.744 *^b^*	0.458 *^b^*
Duration of illness (years)		5.6 ± 4.3	4.8 ± 3.2	1.028 *^b^*	0.306 *^b^*
PHQ-9		15.92 ± 5.73	15.25 ± 5.42	0.619 *^b^*	0.537 *^b^*
HAMD-17		18.87 ± 7.22	18.23 ± 6.39	0.484 *^b^*	0.630 *^b^*
HAMA		17.39 ± 7.75	15.75 ± 6.66	1.151 *^b^*	0.252 *^b^*
SHAPS		32.63 ± 7.40	33.79 ± 5.85	−0.854 *^b^*	0.395 *^b^*
DSST (*n* = 60)		61.30 ± 14.08	61.75 ± 11.74	−0.115 *^b^*	0.909 *^b^*
Maternal warmth		7.09 ± 3.27	14.80 ± 2.74	−12.912 *^b^*	<0.001 *^b^*
Maternal authoritarianism		7.28 ± 2.97	4.58 ± 2.67	4.896 *^b^*	<0.001 *^b^*
Paternal warmth		6.27 ± 4.46	11.4 ± 4.372	−6.036 *^b^*	<0.001 *^b^*
Paternal authoritarianism		6.54 ± 3.07	4.13 ± 2.37	4.394 *^b^*	<0.001 *^b^*

*^a^*: Chi-square value, *^b^*: *t* value.

**Table 2 brainsci-13-00694-t002:** Differences in the right SPL-DMN FC between the AP group and the OP group.

Cluster	Brain Area	L/R	Voxel	MNI Coordinates			*T* Values (Peak)
				*x*	*y*	*z*	
Cluster 1	medial superior frontal cortex	R	13	6	57	3	4.762
Cluster 2	medial superior frontal cortex	R	13	0	60	15	4.092
Cluster 3	medial superior frontal cortex	L	19	−9	51	45	4.069

## Data Availability

The data that support the findings of this study are available from the corresponding author upon reasonable request.

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
