# Peer review of "A Cross-Sectional Study: Structural and Related Functional Connectivity Changes in the Brain: Stigmata of Adverse Parenting in Patients with Major Depressive Disorder?"

_brainsci, 2023, doi:10.3390/brainsci13040694_

Round 1

Reviewer 1 Report

The work attempts to explore the structural and functional changes in the brain of adult major depression disorder patients, as a result of adverse parenting. The patient’s depression severity was identified by various rating scales, and was placed into a parenting group based on appropriate instruments. MRI was then acquired from the patients which showed a change GMV at the right SPL between the adverse parenting and the control group. Also a functional connectivity increase was observed between the right SPL and DMN for the adverse parenting group, compared to the control group. Very weak correlation was seen between parenting subcategories and GMV of the selected ROI. Also, there was no significant correlation between PBI item scores and the depressive symptomes. 

Overall, the manuscript seeks to find the developmental changes in the brain from adverse parenting, leading to depression, through functional imaging. The DMN-FC data although intriguing, does not provide much insight, as such data might be specific to AP subgroups and dominant depression symptoms. The stand-out result is the increased GMV of the right SPL linked with AP. The authors hypothesize that the lack of pruning might be responsible for this seemingly developmental defect. 

A more detailed discussion should be done on the lack of pruning, as it is quite relevant to the finding of this manuscript. 

Also, the administration and effect on depression medication, including diagnosis and treatment history of patients should be clearly stated. 

In figure, 2, the authors should enlarge the brain images, to clearly show the ROI in each, as in the current form they are not easily visible. 

In all figures, images should be clearly labeled.

The parts a, b, c of Figure 2 should be captioned separately.

Reviewer 2 Report

This study reports findings on alterations in grey matter volume and resting state functional connectivity associated with adverse parenting in depression. The investigation was conducted in 125 subjects with a major depressive disorder. Subjects were subdivided into two groups depending on self-reported experiences of aversive parenting (AP) and optimal parenting (OP). 

The manuscript is overall well-written, summarizes important existing research, addresses cultural differences in the concept of adverse parenting, and the study recruited a respectable sample size.

The main weakness of the study is that it lacks a clear hypothesis as a foundation of the performed analyses. The manuscript conveys the impression that an argumentation was developed around an anecdotal finding (group difference in GMV in SPL). The performed structural (and partly also functional) analyses need to be described in more detail. Analyses need to be corrected for multiple comparisons. I suggest the authors describe their original hypothesis in more detail, can then also report if these were violated (e.g. SPL is unusual as a brain region to be expected as affected by AP), and develop their discussion around an unexpected finding.

Please find more specific comments here below:

·      The abstract needs to contain information on sample size and statistics of reported results.

·      The hypotheses at the end of the introduction need to be more specific.

·      The statistical analysis performed on the GMV estimates is insufficiently described.

·      The rational of why a FC analysis of SPL with DMN was performed need to be specified.

- FC analysis needs to be described in more detail: ROI-to-ROI of DMN? How many ROIs were included? were analyses corrected for multiple comparisons? 

·      Sample description: The choice of excluding any comorbidity (also of common ones in depression such anxiety disorders) should be justified.

·      More information on the cluster analysis should be provided, e.g. in the supplemental material. Were cases excluded between the two groups which were very similar to each other?

Author Response

Response to Reviewer 2 Comments

This study reports findings on alterations in grey matter volume and resting state functional connectivity associated with adverse parenting in depression. The investigation was conducted in 125 subjects with a major depressive disorder. Subjects were subdivided into two groups depending on self-reported experiences of aversive parenting (AP) and optimal parenting (OP).

The manuscript is overall well-written, summarizes important existing research, addresses cultural differences in the concept of adverse parenting, and the study recruited a respectable sample size.

The main weakness of the study is that it lacks a clear hypothesis as a foundation of the performed analyses. The manuscript conveys the impression that an argumentation was developed around an anecdotal finding (group difference in GMV in SPL). The performed structural (and partly also functional) analyses need to be described in more detail. Analyses need to be corrected for multiple comparisons. I suggest the authors describe their original hypothesis in more detail, can then also report if these were violated (e.g. SPL is unusual as a brain region to be expected as affected by AP), and develop their discussion around an unexpected finding.

Point 1: The abstract needs to contain information on sample size and statistics of reported results.

Response 1: We gratefully appreciate for your valuable suggestion. We followed this comment and revised the abstracts. We added the following sentence in the Methods of Abstract (Line 21-25, page 1, marked in yellow in manuscript.):

“125 MDD outpatients were included in the study and divided into AP group and optimal parenting (OP) group. Participants completed self-rating scales to assess depressive severity, symptoms and their parents’ styles. They also completed magnetic resonance imaging within one week of filling out instruments. The difference of data was analyzed using the independent samples t-test and chi-squared test.”

Point 2: The hypotheses at the end of the introduction need to be more specific.

Response 2: Thank you for the above suggestions. We have expanded the hypotheses to make it more specific. We added the following sentence at the end of the introduction (Line 94-102, page 1, marked in yellow in manuscript.):

“Here we retrospectively examined the association between AP and multimodal neuroimaging features in patients with MDD participating in the Early Warning Sys-tem and Comprehensive Intervention for Depression (ESCID) project [36,37]. We hypothesized that childhood AP was related to decrease of GMV of amygdala and hip-pocampus and changes inter- or intra- FC of some specific brain networks, such as the DMN, the DAN and the VAN. Our major objectives were to explore possible structural and inter- or intra-network FC changes occurring in MDD patients exposed to AP. First, we measured the GMV within whole brain, and then compared between the optimal parenting (OP) group and AP group. We hypothesized that the differential brain re-gion of GMV was amygdala or hippocampus. Then we calculated the inter- and intra- FC of the DMN, the DAN and the VAN. Our exploratory objectives were to examine relationships between neuroimaging changes and clinical and cognitive features. We estimated the relationship of PBI scores with the values of regional GMV and FC to investigate the effect of parenting style on brain development.

Point 3: The statistical analysis performed on the GMV estimates is insufficiently described.

Response 3: Thank you for pointing out this problem in manuscript. We have expanded the statistical analysis performed on the GMV estimates. We added the following sentence at the end of the introduction (Line 207-210, page 5, marked in yellow in manuscript.):

“ GMV and FC analyses were executed in SPM 8 and SPM 12 separately using a two-sample t-test. To control for confounding effects of age, gender, level of education and duration of illness and head motion parameter, these factors were included in the calculation as covariates. The Gaussian random field (GRF) correction (voxel p-value <0.001, cluster p-value <0.05, one-tailed) was performed to correct for multiple comparisons.”

Point 4: The rational of why a FC analysis of SPL with DMN was performed need to be specified.

Response 4: Thank you for the above suggestions. Thanks for raising this point. We have now added more description of reason why a FC analysis of SPL with DMN was performed in Section 3.3. (Line 229-234, page 6, marked in yellow in manuscript.)

“The differential brain region of GMV analysis between two groups is the right SPL. Although the SPL is not the predicted region of structural change, it is a part of DAN. We hypothesized childhood AP is related to changes in inter-FC of the DAN and the DMN. Consequently, we calculated the FC of SPL with the DMN. In the DMN, voxel-wise FC between the right SPL and the right smFC (Cluster 1), the right smFC (Cluster 2) and the left smFC (Cluster 3) were significantly different between AP group and OP group (Table 2).”

Point 5: FC analysis needs to be described in more detail: ROI-to-ROI of DMN? How many ROIs were included? were analyses corrected for multiple comparisons?

Response 5: We are very sorry for our negligence of the description. All the FC analysises were voxel-wise FC analysises. There were two ROIs included. ROI 1 was the brain region in which GMV differed significantly between the AP group and OP group. And to increase the robustness of the voxel-wise FC results, we extracted a sphere with a radius of 6 mm(ROI 2) that centered on peak MNI coordinates of ROI 1 and performed a sensitive analysis (see S2 of Supplementary Materials). All analyses corrected for multiple comparisons using the Gaussian random field (GRF) correction (voxel p-value <0.001, cluster p-value <0.05, one-tailed). In Section 2.5 and 2.6, we have provided detailed descriptions of FC analysis at Section 2.5 and 2.6.( Line 189-192 and 209-210, page 5, marked in yellow in manuscript.)

2.5. Voxel-wise FC analyses

Wake Forest University (WFU) PickAtlas 3.0.5b [41] was used to obtain region of interest (ROI) templates in the MNI space. FC analysis was completed using RESTplus v1.2 after data preprocessing. Brain region in which GMV differed significantly between the AP group and OP group was selected as seed points (ROI 1) in the voxel-wise FC analysis. To increase the robustness of the FC results, we performed sensitivity analysis as follows: a sphere with a radius of 6 mm centered on coordinates of ROI 1 was selected as the region of interest (ROI 2) using WFU PickAtlas 3.0.5b. The average time series of all the voxels in ROIs 1 and 2 were extracted and calculated. Then, voxel-wised Pearson’s correlation coefficients were calculated on the time series between ROI 1/2 and DMN [42] and then converted to Z values by Fisher’s equation to generate subject-specific maps [43]. Thus, the Z value representation of the functional connections between ROIs with the DMN were obtained. We used the same approach to analyze the intra-network FCs.”

2.6. Statistical analyses

        Differences in sex and education levels between the AP and OP groups were assessed by chi-squared analysis. The independent samples t-test was applied to explore between-group differences in age, duration of illness, and PBI, PHQ-9, and HAMD-17 scores. Pearson’s correlation coefficients were used to determine correlations between GMVs and PBI item scores. A p-value of <0.05 (two-tailed) was deemed statistically significant. All analyses were conducted in IBM SPSS Statistics v25.0 (IBM Statistics, Armonk, NY, USA). GMV and FC analyses were executed in SPM 8 and SPM 12 separately using a two-sample t-test. To control for confounding effects of age, gender, level of education and duration of illness and head motion parameter, these factors were included in the calculation as covariates. The Gaussian random field (GRF) correction (voxel p-value <0.001, cluster p-value <0.05, one-tailed) was performed to correct for multiple comparisons. ”

Point 6: Sample description: The choice of excluding any comorbidity (also of common ones in depression such anxiety disorders) should be justified.

Response 6: Thank you for pointing out this problem in manuscript. Indeed, comorbidity is a very important contributing factor and it’s almost impossible to completely exclude MDD without symptoms of anxiety. The severity of anxiety symptoms in the included patients with MDD did not meet diagnostic criteria for anxiety according to DSM-5. And there is no significant difference in HAMA between the two groups.

" Exclusion criteria were: 1) mental health disorders other than MDD diagnosed according to DSM-5; 2) a history of organic brain disease; 3) severe stupor or other symptoms that could intervene with the study; 4) transcranial magnetic stimulation (TMS) or electroconvulsive therapy (ECT) treatment within the prior six months; and 5) pregnancy. "

Table 1. Demographics and clinical characteristics.

Characteristics (mean + SD)

Adverse parenting

Optimal parenting

χ2/t

P

n (%)

n (%)

Gender

Female

67 (78.8%)

26 (65.0%) 

2.729a

0.099 a

Male

18 (21.2%)

14 (35.0%)

Education level

High school or

6 (7.1%)

4 (10%)

0.339a

0.844 a

less

Undergraduate

65 (76.4%)

30 (75%)

Postgraduate or

14 (16.5%)

6 (15%)

higher

Medications

Drug-naïve

50(58.8%)

23(57.5%)

0.020 a

0.889 a

Age (years)

25.6 + 6.2

24.7 + 5.4

0.744b

0.458 b

Duration of illness (years)

5.6 + 4.3

4.8 + 3.2

1.028b

0.306 b

PHQ-9

15.92 + 5.73

15.25 + 5.42

0.619b

0.537b

HAMD-17 

18.87 + 7.22

18.23 + 6.39

0.484b

0.630b

HAMA

17.39 + 7.75

15.75 + 6.66

1.151 b

0.252 b

SHAPS

32.63 + 7.40

33.79 + 5.85

-0.854b

0.395b

DSST (n = 60)

61.30 + 14.08

61.75 + 11.74

-0.115b

0.909b

Maternal warmth

7.09 + 3.27

14.80 + 2.74

-12.912b

<0.001b

Maternal authoritarianism

7.28 + 2.97

4.58 + 2.67

4.896b

<0.001b

Paternal warmth

6.27 + 4.46

11.4 + 4.372

-6.036b

<0.001b

Paternal authoritarianism

6.54 + 3.07

4.13 + 2.37

4.394b

<0.001b

a: Chi-square value, b: t value.”

Point 7: More information on the cluster analysis should be provided, e.g. in the supplemental material. Were cases excluded between the two groups which were very similar to each other?

Response 7: Thank you for the above suggestions. We have added the information on the cluster analysis in Section S1 of supplemental material. There were no cases that very similar belong to different groups, this may because the trend the Warmth factor score and the Authoritarianism factor is basically opposite in our sample. For this reason, we kept all the cases. We added the following sentence in the S1 of Supplementary Material:

S1. Cluster analysis

We selected 4 variables (maternal Warmth factor score, maternal Authoritarianism factor score, paternal Warmth factor score and paternal Authoritarianism factor score) of 125 participants with which we performed the Ward minimum-variance hierarchical clustering using a Euclidean distance metric with complete linkage. The analysis was conducted in IBM SPSS Statistics v25.0 (IBM Statistics, Armonk, NY, USA).”

Reviewer 3 Report

In this study, the Authors provided new evidence that adverse parenting (AT) manifesting in low warmth, ignorance and high authoritarianism parenting style, during childhood is related to depressive disorders and changes in brain anatomy. Based on the obtained results, the Authors suggested that brain changes may be the imprint of the childhood adverse parenting that affects certain depressive symptoms. In other words, the Authors provided evidence that AP may be a serious factor of developmental programming of adult brain disorders.

This is an excellent article with bright main idea. There are several small points:

1.     Abstract. 

Please, provide clear aim of the study and if it is possible, please, remove limitations of the study from the Abstract. It also would be good to provide several words about the application of the study.

2.     Discussion.

If it is possible, please, discuss possible molecular mechanisms of the connection between AP and long-term major depression disorders. Just in several lines. For example, it is well established that increased blood cortisol level induced by AP can increase inflammation and disturb immunity thereby leading to the alteration of the proliferation/differentiation balance in the young maturating brain.

If it is possible, provide a small chapter describing future directions of the study and their potential applications.

Author Response

Response to Reviewer 3 Comments

In this study, the Authors provided new evidence that adverse parenting (AP) manifesting in low warmth, ignorance and high authoritarianism parenting style, during childhood is related to depressive disorders and changes in brain anatomy. Based on the obtained results, the Authors suggested that brain changes may be the imprint of the childhood adverse parenting that affects certain depressive symptoms. In other words, the Authors provided evidence that AP may be a serious factor of developmental programming of adult brain disorders.

Point 1: Abstract.

Please, provide clear aim of the study and if it is possible, please, remove limitations of the study from the Abstract. It also would be good to provide several words about the application of the study.

Response 1: We gratefully appreciate for your valuable suggestions. We have revised the Abstract as your suggestions. Yet the limitation seems to be an indispensable part of the Abstract, for this reason we failed to revome it.

Abstract: Background: There is a high correlation between risk of major depression disorder (MDD) and adverse childhood experiences (ACEs) like adverse parenting (AP). While there appears to be an association between ACEs and changes in brain structure and function, there have yet to be multimodal neuroimaging studies of associations between parenting style and brain developmental changes in MDD patients. To explore the effect of AP on brain structure and function. Methods: In this cross-sectional study, 125 MDD outpatients were included in the study and divided into AP group and optimal parenting (OP) group. Participants completed self-rating scales to assess depressive severity, symptoms and their parents’ styles. They also completed magnetic resonance imaging within one week of filling out instruments. The difference of data was analyzed using the independent samples t-test and chi-squared test. Differences in gray matter volume (GMV) and resting-state functional connectivity (RS-FC) were assessed between groups. Results: AP was associated with a significant increase in GMV in the right superior parietal lobule (SPL) and FC between the right SPL and the bilateral medial superior frontal cortex in MDD patients. Limitations: The cross-cultural characteristics of AP will result in the lack of the generalizability of the findings. Conclusions: The results support the hypothesis that AP during childhood may imprint the brain to affect depressive symptoms in adulthood. Parents should pay attention to the parenting style, avoid the use of parenting that lacks of warmth.”

Point 2: Discussion.

If it is possible, please, discuss possible molecular mechanisms of the connection between AP and long-term major depression disorders. Just in several lines. For example, it is well established that increased blood cortisol level induced by AP can increase inflammation and disturb immunity thereby leading to the alteration of the proliferation/differentiation balance in the young maturating brain.

Response 2: We gratefully thanks for the precious time the reviewer spent making constructive remarks. We have added discussion about possible molecular mechanisms of the connection between AP and long-term major depression disorders in Section 4.2(Line 312-316, page 9, marked in yellow in manuscript.):

“We found that markedly increased GMV in the right SPL was associated with AP. Previous studies have suggested that AP experiences during childhood and adolescence generate chronic hypothalamus-pituitary-adrenal (HPA) axis hyperactivity, resulting in abnormal neurodevelopment in the normal (non-depressed) population; for example, reduced GMV in the hippocampus [59,60]. Exposure to AP could activate the release of corticotropin-releasing hormone (CRH) and induce increased production of glucocorticoids from the HPA-axis and higher levels of proinflammatory cytokines in the brain[61,62]. The glucocorticoids and proinflammatory cytokines are related to brain abnormalities in depression[63].”

Point 3: If it is possible, provide a small chapter describing future directions of the study and their potential applications.

Response 3: Thank you for the above suggestion. Following your suggestion, we have added some contents of future directions and potential applications in Section 5(Line 406-409, page 11, marked in yellow in manuscript.):

“ Here we found that AP, as a low warmth and high authoritarianism parenting style, during childhood is related to significant enlargements in the right SPL and increased FC between the right SPL and the bilateral smFC in MDD patients. These structural and functional changes were also negatively correlated with AP characteristics and the clinical features of anxiety/somatization. A more well-designed longitudinal study with a larger sample from different cultural background is essential to prove the effect of AP. The findings further emphasis the importance of optimal and appropriate parenting and arouse the attention of society and families to the parenting style. Our study generates an intriguing hypothesis that brain changes may be the imprint of the childhood AP that affect certain depressive symptoms.”

Reviewer 4 Report

Little data exist concerning the relationship of parenting styles to the development of depression later in life. In particular, there is a serious underrepresentation of studies of adverse parenting. As such, this work offers potentially important new insights for possible prevention and treatment of this psychiatric disorder.

This is a fairly (see below) well-written account of well-designed and well-executed structural and functional brain imaging studies in a Chinese population. Subjects were chosen based on having a diagnosis of Major Depressive Disorder (MDD). They were then subdivided into those whose parents displayed low warmth and high authoritarian tendencies (AP: adverse parenting) vs. those with MDD whose parents displayed high warmth and low authoritarian styles (OP: optimal parenting. One minor flaw is that the results are not immediately generalizable to Western populations due to cultural differences in how parenting styles are perceived by offspring and societies.  However, the authors rightly acknowledge this and propose resolving the cultural interpretation as fodder for future work.

The results are relatively modest: structural analysis revealed AP was associated with increased gray matter volume in the right superior parietal lobule. Subsequent analysis of resting state functional connectivity (rs-fc) linked this region bilaterally to the medial superior frontal cortex.

Lines 228-229. This is where it gets harder to follow. The authors then go on to examine the relationship between parenting style and the functional connectivity of the default mode network (DMN) and the dorsal attention network (DAN). Why? The reason for this was not immediately clear to this reader: apparently based on the literature cited between lines 62-72 (or 325-344?).  In any event they then found a relationship between the left medial superior frontal cortex and the DMN.

The well-cited Discussion plausibly links these findings to previously reported changes in the DMN, attentional shifting and other networks illuminated by functional imaging work.

Overall, this is a very good offering that just needs some polish. The most serious drawback with this manuscript, as written, is that it is only immediately understandable to those readers thoroughly versed in the structural/functional imaging literature.  The scope of the journal is stated as being intended for a broad audience of brain scientists. For those studying other aspects of neuroscience, a lot of homework (PubMed, Scopus, Dr. Google and Dr. Wiki) must be done in order to understand what was done and why. Assuming the authors (and editors) would like to communicate with a broader audience, the Introduction would benefit by adding a paragraph expanding the descriptions of attention shifting networks. Specifically, the Intro needs to explain the imaging concept of re-fc with a clear statement early in the paper that rs-fc does not actually reflect direct underlying neural connections, but rather temporally related activity patterns. You might add that, although met with some initial skepticism, the existence of these “circuits” has stood up to widespread experimental work over many years and explains well-known psychiatric/cognitive phenomena. Pease define what the pathway acronyms stand for (VAN, DAN, DMN (dorsomedial thalamic nucleus?) and a brief description of the pathway function as currently understood and how it relates to other pathways in the paper.  Finally, which pathways are relevant to the current investigation and why?

The Figures are generally very nice. The mid-sagittal frontal cortex ROI in 2C does not show up very well.

Author Response

Response to Reviewer 4 Comments

Little data exist concerning the relationship of parenting styles to the development of depression later in life. In particular, there is a serious underrepresentation of studies of adverse parenting. As such, this work offers potentially important new insights for possible prevention and treatment of this psychiatric disorder.

This is a fairly (see below) well-written account of well-designed and well-executed structural and functional brain imaging studies in a Chinese population. Subjects were chosen based on having a diagnosis of Major Depressive Disorder (MDD). They were then subdivided into those whose parents displayed low warmth and high authoritarian tendencies (AP: adverse parenting) vs. those with MDD whose parents displayed high warmth and low authoritarian styles (OP: optimal parenting. One minor flaw is that the results are not immediately generalizable to Western populations due to cultural differences in how parenting styles are perceived by offspring and societies.  However, the authors rightly acknowledge this and propose resolving the cultural interpretation as fodder for future work.

The results are relatively modest: structural analysis revealed AP was associated with increased gray matter volume in the right superior parietal lobule. Subsequent analysis of resting state functional connectivity (rs-fc) linked this region bilaterally to the medial superior frontal cortex.

Point 1: Lines 228-229. This is where it gets harder to follow. The authors then go on to examine the relationship between parenting style and the functional connectivity of the default mode network (DMN) and the dorsal attention network (DAN). Why? The reason for this was not immediately clear to this reader: apparently based on the literature cited between lines 62-72 (or 325-344?).  In any event they then found a relationship between the left medial superior frontal cortex and the DMN.

Response 1: Thank you very much for pointing out this problem in manuscript. We have provided a clear reason of analysis of functional connectivity between the DMN and the DAN in Section 3.3(Line 229-232, page 6, marked in yellow in manuscript.):

“The differential brain region of GMV analysis between two groups is the right SPL. Although the SPL is not the predicted region of structural change, it is a part of DAN. We hypothesized childhood AP is related to changes in inter-FC of the DAN and the DMN. Consequently, we calculated the FC between the SPL and the DMN.”

The well-cited Discussion plausibly links these findings to previously reported changes in the DMN, attentional shifting and other networks illuminated by functional imaging work.

Point 2: Overall, this is a very good offering that just needs some polish. The most serious drawback with this manuscript, as written, is that it is only immediately understandable to those readers thoroughly versed in the structural/functional imaging literature.  The scope of the journal is stated as being intended for a broad audience of brain scientists. For those studying other aspects of neuroscience, a lot of homework (PubMed, Scopus, Dr. Google and Dr. Wiki) must be done in order to understand what was done and why. Assuming the authors (and editors) would like to communicate with a broader audience, the Introduction would benefit by adding a paragraph expanding the descriptions of attention shifting networks. Specifically, the Intro needs to explain the imaging concept of rs-fc with a clear statement early in the paper that rs-fc does not actually reflect direct underlying neural connections, but rather temporally related activity patterns. You might add that, although met with some initial skepticism, the existence of these “circuits” has stood up to widespread experimental work over many years and explains well-known psychiatric/cognitive phenomena. Please define what the pathway acronyms stand for (VAN, DAN, DMN (dorsomedial thalamic nucleus?) and a brief description of the pathway function as currently understood and how it relates to other pathways in the paper.  Finally, which pathways are relevant to the current investigation and why?

Response 2: We gratefully appreciate for your valuable suggestion. The full names of the networks(VAN, DAN, DMN) have been defined where they first appeared. The rest of the responses are placed in the Section 1(Line 62-69 and 77-81, page 2, marked in yellow in manuscript.):

 “In addition to affecting brain structure, ACEs might also affect brain function and resting-state functional connectivity (RS-FC). RS-FC is an index that measures the degree of temporally synchrony of the blood oxygenation level dependent signal among different brain regions. Yeo et al. organized a 7-network parcellation based on distinct characteristic of different cortical regions and data from 1000 subjects[23]. Despite RS-FC raised some initial skepticism, it has withstood extensive experimental work over the years. RS-FC can not only explain some behavioral outputs such as sustained attention, severity of psychiatric symptoms, and personality traits but can also differentiate subjects with accuracy[24-26]. For example, childhood neglect is related to weaker amygdala RS-FC, with clusters detected within the left dorsal precuneus [27] and increased RS-FC between the salience network (SN) and the default mode network (DMN) [28]. Early-life stress is positively associated with within-network connectivity of the ventral attention network (VAN), the dorsal attention network (DAN), and the between-network connectivity of the VAN-DAN [18]. Both ACEs and internalizing symptoms were related to a decreased anticorrelation between the DMN and the DAN [29], and positive parenting was associated with decreased coactivation of the superior parietal lobule (SPL), a component of the DAN, with the executive control network [29]. The DMN plays a vital role in cognitive and social functions, such as self-referential processes and the FC within-DMN is associated with positive parenting traits [30,31]. The DAN and the VAN are specialized for disparate subprocesses of attention but tend to cooperate to remain normal cognitive processes [32]. The RS-FC within and between the VAN and the DAN is related to early life stress [18].”

Point 3: The Figures are generally very nice. The mid-sagittal frontal cortex ROI in 2C does not show up very well.

Response 3: Thank you so much for your careful check. We followed this comment and revised the and enlarge the brain images in figure 3(We added a flowchart as Figure 1, so the serial number of the figure is extended by one) (Line 243, page 7).

Reviewer 5 Report

Thank you for giving me the opportunity to review this manuscript.

1) Please attach the STROBE checklist and fill in the page numbers.

2) Please describe the study design clearly in the title and the abstract. I think this study is a cross-sectional study.

3) Please descrive the setting and the relevant dates, including periods of exposure, follow-up and data collection.

4) Please clearly define all predictors, potential confounders,and effect modifiers. Please describe all statistical methods including those used to control for confoundings. Please describe unadjustedn estimates and, if applicable, confounder-adjusted estimates and their precisions.

5) Please explain how sample size was arrived at, and how missing data were addressed. Please describe how much sample size was warranted to improve this study well enough.

6) In the baseline characteristics, DSST scores, maternal warmth, maternal authoritarianism, paternal warmth, and paternal authoritarianism were significantly different between adverse parenting group and optimal parenting group. Please describe how much attention and working memory,  affect the right SPL and the bilateral SMFC in this study (the cross-sectional design)?

7) In this study, the structural and functional changes were also negatively correlated with AP characteristics and the clinical features of anxiety/somatization. Then, please explain how much anxiety/somatization in the baseline affect the right SPL and the bilateral SMFC. Anxiety and Somatization were possibly due to nomal anxiety because of participating in this study. I think Anxiety and Somatization were not always related to depression.

I think it is necessary to revise the manuscript.

Author Response

Response to Reviewer 5 Comments

Point 1: Please attach the STROBE checklist and fill in the page numbers.

Response 1: Thank for your suggestion. The STROBE checklist is now attached.

STROBE Statement—checklist of items that should be included in reports of observational studies

Item No

Recommendation

Page
No

Title and abstract

1

(a) Indicate the study’s design with a commonly used term in the title or the abstract

1

(b) Provide in the abstract an informative and balanced summary of what was done and what was found

1

Introduction

Background/rationale

2

Explain the scientific background and rationale for the investigation being reported

1-3

Objectives

3

State specific objectives, including any prespecified hypotheses

2-3

Methods

Study design

4

Present key elements of study design early in the paper

3-6

Setting

5

Describe the setting, locations, and relevant dates, including periods of recruitment, exposure, follow-up, and data collection

3-4

Participants

6

(a) Cohort study—Give the eligibility criteria, and the sources and methods of selection of participants. Describe methods of follow-up

Case-control study—Give the eligibility criteria, and the sources and methods of case ascertainment and control selection. Give the rationale for the choice of cases and controls

Cross-sectional study—Give the eligibility criteria, and the sources and methods of selection of participants

3

(b) Cohort study—For matched studies, give matching criteria and number of exposed and unexposed

Case-control study—For matched studies, give matching criteria and the number of controls per case

n/a

Variables

7

Clearly define all outcomes, exposures, predictors, potential confounders, and effect modifiers. Give diagnostic criteria, if applicable

3-5

Data sources/ measurement

8*

 For each variable of interest, give sources of data and details of methods of assessment (measurement). Describe comparability of assessment methods if there is more than one group

3-5

Bias

9

Describe any efforts to address potential sources of bias

n/a

Study size

10

Explain how the study size was arrived at

3

Quantitative variables

11

Explain how quantitative variables were handled in the analyses. If applicable, describe which groupings were chosen and why

3-4

Statistical methods

12

(a) Describe all statistical methods, including those used to control for confounding

4-5

(b) Describe any methods used to examine subgroups and interactions

n/a

(c) Explain how missing data were addressed

n/a

(d) Cohort study—If applicable, explain how loss to follow-up was addressed

Case-control study—If applicable, explain how matching of cases and controls was addressed

Cross-sectional study—If applicable, describe analytical methods taking account of sampling strategy

n/a

Loss to follow-up was one of our outcomes observed

(e) Describe any sensitivity analyses

4

Continued on next page

Results

Participants

13*

(a) Report numbers of individuals at each stage of study—eg numbers potentially eligible, examined for eligibility, confirmed eligible, included in the study, completing follow-up, and analysed

3

(b) Give reasons for non-participation at each stage

n/a

(c) Consider use of a flow diagram

3, Fig 1

Descriptive data

14*

(a) Give characteristics of study participants (eg demographic, clinical, social) and information on exposures and potential confounders

5, Table 1

(b) Indicate number of participants with missing data for each variable of interest

n/a

(c) Cohort study—Summarise follow-up time (eg, average and total amount)

n/a

Outcome data

15*

Cohort study—Report numbers of outcome events or summary measures over time

n/a

Case-control study—Report numbers in each exposure category, or summary measures of exposure

n/a

Cross-sectional study—Report numbers of outcome events or summary measures

5, Table 1

Main results

16

(a) Give unadjusted estimates and, if applicable, confounder-adjusted estimates and their precision (eg, 95% confidence interval). Make clear which confounders were adjusted for and why they were included

n/a

(b) Report category boundaries when continuous variables were categorized

Table 1

(c) If relevant, consider translating estimates of relative risk into absolute risk for a meaningful time period

Not part of our analysis

Other analyses

17

Report other analyses done—eg analyses of subgroups and interactions, and sensitivity analyses

S2 of Supplementary Materials

Discussion

Key results

18

Summarise key results with reference to study objectives

9-11

Limitations

19

Discuss limitations of the study, taking into account sources of potential bias or imprecision. Discuss both direction and magnitude of any potential bias

11

Interpretation

20

Give a cautious overall interpretation of results considering objectives, limitations, multiplicity of analyses, results from similar studies, and other relevant evidence

9-11

Generalisability

21

Discuss the generalisability (external validity) of the study results

11

Other information

Funding

22

Give the source of funding and the role of the funders for the present study and, if applicable, for the original study on which the present article is based

11

*Give information separately for cases and controls in case-control studies and, if applicable, for exposed and unexposed groups in cohort and cross-sectional studies.

Note: An Explanation and Elaboration article discusses each checklist item and gives methodological background and published examples of transparent reporting. The STROBE checklist is best used in conjunction with this article (freely available on the Web sites of PLoS Medicine at http://www.plosmedicine.org/, Annals of Internal Medicine at http://www.annals.org/, and Epidemiology at http://www.epidem.com/). Information on the STROBE Initiative is available at www.strobe-statement.org.

Point 2: Please describe the study design clearly in the title and the abstract. I think this study is a cross-sectional study.

Response 2: Thank you for the above suggestions. We have revised the title and the abstract.

Title: A cross-sectional study: Structural and related functional connectivity changes in the brain: stigmata of adverse parenting in patients with major depressive disorder?

Abstract: Methods: In this cross-sectional study, 125 MDD outpatients were included in the study and divided into AP group and optimal parenting (OP) group.”

Point 3: Please describe the setting and the relevant dates, including periods of exposure, follow-up and data collection.

Response 3: Thank for your suggestions. This is a cross-sectional study, hence we didn’t follow-up the participants. The data collection of the study were has been described in Section 2.1 and Figure 1.( Line 107-126, page 3.)

2.1. Study design, participants, and inclusion and exclusion criteria

This observational study was undertaken at the Renmin Hospital of Wuhan University following the principles of the Declaration of Helsinki [38]. The Ethics Committee of Renmin Hospital of Wuhan University, Wuhan, Hubei, China approved the study protocol. Written informed consent was obtained from all participants. This report was written following the STROBE statement [39].

MDD patients were selected from the Early Warning System and Comprehensive Intervention for Depression (ESCID) project. We calculated the sample size using G*Power software (latest ver. 3.1.9.7; Heinrich-Heine-Universität Düsseldorf, Düsseldorf, Germany). Informed consents of 8 participants were withdrawn and 11 participants didn’t meet the inclusion criteria. One hundred and twenty-five MDD outpatients were recruited between May 2019 and April 2022. The inclusion criteria were: 1) aged 18–50 years; 2) meeting DSM-5 diagnostic criteria for MDD diagnosed by two experienced psychiatrists; and 3) having a junior high school education or higher. Exclusion criteria were: 1) mental health disorders other than MDD diagnosed according to DSM-5; 2) a history of organic brain disease; 3) severe stupor or other symptoms that could intervene with the study; 4) transcranial magnetic stimulation (TMS) or electroconvulsive therapy (ECT) treatment within the prior six months; and 5) pregnancy. (Figure 1)

Figure 1. Flowchart representing procedure of study design and participants.”

Point 4: Please clearly define all predictors, potential confounders, and effect modifiers. Please describe all statistical methods including those used to control for confoundings. Please describe unadjusztedn estimates and, if applicable, confounder-adjusted estimates and their precisions.

Response 4: Thank you for pointing out this problem in statistical analyses. We have revised the insufficiency of Section 2.6. We added the following sentence in the Section 4.2 (Line 315-319, page 10, marked in yellow in manuscript.):

“2.6. Statistical analyses

Differences in sex and education levels between the AP and OP groups were as-sessed by chi-squared analysis. The independent samples t-test was applied to explore between-group differences in age, duration of illness, and PBI, PHQ-9, and HAMD-17 scores. Pearson’s correlation coefficients were used to determine correlations between GMVs and PBI item scores. A p-value of <0.05 (two-tailed) was deemed statistically significant. All analyses were conducted in IBM SPSS Statistics v25.0 (IBM Statistics, Armonk, NY, USA). GMV and FC analyses were executed in SPM 8 and SPM 12 sepa-rately using a two-sample t-test. To control for confounding effects of age, gender, level of education and duration of illness and head motion parameter, these factors were included in the calculation as covariates. The Gaussian random field (GRF) correction (voxel p-value <0.001, cluster p-value <0.05, one-tailed) was performed to correct for multiple comparisons.”

Point 5: Please explain how sample size was arrived at, and how missing data were addressed. Please describe how much sample size was warranted to improve this study well enough.

Response 5: Thank for your suggstions. We calculated the sample size using G*Power software (latest ver. 3.1.9.7; Heinrich-Heine-Universität Düsseldorf, Düsseldorf, Germany) with the effect size of 0.5, the default significance level of 0.05 and a power of 0.8. The total sample size is 128, so we recruited 144 participants. However, informed consents of 8 participants were withdrawn and 11 participants didn’t meet the inclusion criteria. The study included 125 samples at last. Because great heterogeneity of fMRI study and the cross-cultural characteristics of AP, a larger sample from different cultural background is essential to verify the effect of AP in the future. We added the following sentence in the Section 4.2 (Line 315-319, page 10, marked in yellow in manuscript.):

“MDD patients were selected from the Early Warning System and Comprehensive Intervention for Depression (ESCID) project. We calculated the sample size using G*Power software (latest ver. 3.1.9.7; Heinrich-Heine-Universität Düsseldorf, Düsseldorf, Germany). Informed consents of 8 participants were withdrawn and 11 participants didn’t meet the inclusion criteria. One hundred and twenty-five MDD outpatients were recruited between May 2019 and April 2022. The inclusion criteria were: 1) aged 18–50 years; 2) meeting DSM-5 diagnostic criteria for MDD diagnosed by two experienced psychiatrists; and 3) having a junior high school education or higher. Exclusion criteria were: 1) mental health disorders other than MDD diagnosed according to DSM-5; 2) a history of organic brain disease; 3) severe stupor or other symptoms that could intervene with the study; 4) transcranial magnetic stimulation (TMS) or electroconvulsive therapy (ECT) treatment within the prior six months; and 5) pregnancy. (Figure 1).

Figure 1. Flowchart representing procedure of study design and participants.”

Point 6: In the baseline characteristics, DSST scores, maternal warmth, maternal authoritarianism, paternal warmth, and paternal authoritarianism were significantly different between adverse parenting group and optimal parenting group. Please describe how much attention and working memory,  affect the right SPL and the bilateral SMFC in this study (the cross-sectional design)?

Response 6: We gratefully appreciate for your valuable comment. However, “There were no significant differences in age, gender, education level, medications, HAMD-17, PHQ-9, HAMA, SHAPS, DSST, or duration of illness between the two groups.” (Line 215-216, page 5, marked in yellow in manuscript.) We are very sorry for our unclear expression.

Point 7: In this study, the structural and functional changes were also negatively correlated with AP characteristics and the clinical features of anxiety/somatization. Then, please explain how much anxiety/somatization in the baseline affect the right SPL and the bilateral SMFC. Anxiety and Somatization were possibly due to nomal anxiety because of participating in this study. I think Anxiety and Somatization were not always related to depression.

Response 7: We gratefully appreciate for your valuable comment. In the study, AP characteristics are correlated with the structural and functional changes and the clinical features of anxiety/somatization. While there was no correlation between the structural and functional changes and the clinical features of anxiety/somatization. We are very sorry for our unclear expression.

“As shown in Figure 4, the GMV of the ROI was significantly negatively correlated with maternal warmth scores (r = -0.329, p < 0.001) and positively correlated with maternal authoritarianism scores (r = 0.196, p = 0.037) and paternal authoritarianism scores (r = 0.195, p = 0.038).” (Line 264-267, page 8, marked in yellow in manuscript.)

Figure 4. A negative correlation between the GMV of the ROI and maternal warmth scores (r = -0.329, p < 0.001) and a positive correlation between the GMV of the ROI and parental authoritarianism scores (for father: r = 0.196, p = 0.037; for mother: r = 0.195, p = 0.038) were observed.

” There was a positive correlation between paternal authoritarianism and anxiety/somatization scores, but no significant results in the other symptoms.” (Line 281-283, page 9, marked in yellow in manuscript.) We are very sorry for our unclear expression.

Table 1. Pearson correlation between PBI item scores and FC signals.

Maternal Warmth

Maternal Authoritarianism

Paternal Warmth

Paternal Authoritarianism

FC signal of Cluster 1

-0.368***

0.139

-0.232*

0.010

FC signal of Cluster 2

-0.264**

0.021

-0.255**

0.047

FC signal of Cluster 3

-0.368***

0.033

-0.207*

0.013

FC signal of Cluster 4

-0.504***

0.251**

-0.219*

0.145

*P < 0.05, **P < 0.01, ***P < 0.001 (2-tailed)

Cluster 1, Cluster 2 and Cluster 3 are the significantly different regions in which that the voxel-wised FC between the right SPL and the DMN. Cluster 4 is the significantly different region in which that the voxel-wised FC between the ROI 3 and the DMN.

Round 2

Reviewer 2 Report

The authors have put significant effort into revising their manuscript. They have addressed most of my comments. Only a few minor points remain that need to be revised:

Point 1

-        In the abstract: Please specify which group differences in what data were tested with which statistical tests. 

Point 3

-        Controlling for covariates in a t-test is not possible: Isn’t it a regression analysis that you performed?

Point 5:

-        Please provide information on the GRF correction method and a reference for this correction method. I have never heard of this correction method and couldn't find any information on a quick google search. 

Additional point:

-        Please provide information on the overall number of tests performed on the structural and functional data set. The overall number of tests performed needs to be taken into account by adjusting for multiple comparisons (e.g. with a Bonferroni correction) overall. With so many tests performed on one data set, to correct for multiple comparisons is crucial

Author Response

Response to Reviewer 2 Comments

The authors have put significant effort into revising their manuscript. They have addressed most of my comments. Only a few minor points remain that need to be revised:

Point 1: In the abstract: Please specify which group differences in what data were tested with which statistical tests.

Response 1: Thank you so much for your careful check. We now revised the following sentence in the abstract (Line 24-26, page 1, marked in yellow in manuscript.):

Methods: In this cross-sectional study, 125 MDD outpatients were included in the study and divided into AP group and optimal parenting (OP) group. Participants completed self-rating scales to assess depressive severity, symptoms and their parents’ styles. They also completed magnetic resonance imaging within one week of filling out instruments. The differences between groups of gender, educational level and medications were analyzed using chi-squared test and that of age, duration of illness and scores of scales using the independent samples t-test. Differences in gray matter volume (GMV) and resting-state functional connectivity (RS-FC) were assessed between groups.”

Point 2: Controlling for covariates in a t-test is not possible: Isn’t it a regression analysis that you performed?

Response 2: We are very sorry for our negligence of the statistical analyses. The statistical analysises have used the general linear model (GLM) which is an extension of simple linear regression. The design compared between two groups displayed as “Two-sample t-test” in the interface. So we failed to describe the accurate the statistical process. We revised the following sentence in the Section 2.6 (Line 207-208, page 5, marked in yellow in manuscript.):

“GMV and FC analyses were executed in SPM 8 and SPM 12 separately using the general linear model.”

The GLM allows statistical testing of various components and production of parametric activation maps. In short, the use of the GLM approach could separate stimulus induced signals from noise. Statistical parametric maps (SPMs) are image processes with voxel values that are, under the null hypothesis, distributed according to a known probability density function, usually the Student's T or F distributions. So they are valid providing the residuals, after fitting the model, are independent and normally distributed[1].

Point 3: Please provide information on the GRF correction method and a reference for this correction method. I have never heard of this correction method and couldn't find any information on a quick google search.

Response 3: We feel sorry for the inconvenience brought to the reviewer. Gaussian random field (GRF) correction is extensively used in correct for multiple comparisons in statistical analysis of medical images since GRF theory was arised[2-6]. And we have added references for this correction method in the manuscript (Line 212, page 5, marked in yellow in manuscript.).

Additional point: Please provide information on the overall number of tests performed on the structural and functional data set. The overall number of tests performed needs to be taken into account by adjusting for multiple comparisons (e.g. with a Bonferroni correction) overall. With so many tests performed on one data set, to correct for multiple comparisons is crucial.

Response 4: We totally understand the reviewer's concern. There were several hundred thousand voxels participated in the statistical test in the voxel-based morphometry that analyzed the whole brain. So it is necessary to correct for these multiple dependent comparisons as your above commention. And we have performed the significance threshold correction according to Gaussian random field (GRF) theory with a voxel level of P < 0.001 and a cluster level of P < 0.05 using the RESTplus v1.2 toolbox. The GRF is a reliable correction method for controlling false positives rates.

Reference:

  1. Ashburner, J.; Friston, K.J. Voxel-based morphometry--the methods. Neuroimage 2000, 11, 805-821, doi:10.1006/nimg.2000.0582.
  2. Friston, K.J.; Frith, C.D.; Liddle, P.F.; Frackowiak, R.S. Comparing functional (PET) images: the assessment of significant change. J Cereb Blood Flow Metab 1991, 11, 690-699, doi:10.1038/jcbfm.1991.122.
  3. Friston, K.J.; Holmes, A.; Poline, J.B.; Price, C.J.; Frith, C.D. Detecting activations in PET and fMRI: levels of inference and power. Neuroimage 1996, 4, 223-235, doi:10.1006/nimg.1996.0074.
  4. Nichols, T.E. Multiple testing corrections, nonparametric methods, and random field theory. Neuroimage 2012, 62, 811-815, doi:10.1016/j.neuroimage.2012.04.014.
  5. Worsley, K.J.; Taylor, J.E.; Tomaiuolo, F.; Lerch, J. Unified univariate and multivariate random field theory. Neuroimage 2004, 23 Suppl 1, S189-195, doi:10.1016/j.neuroimage.2004.07.026.
  6. Cheng, H.; Zhang, Z.; Zhang, B.; Zhang, W.; Wang, J.; Ni, W.; Miao, Y.; Liu, J.; Bi, Y. Enhancement of Impaired Olfactory Neural Activation and Cognitive Capacity by Liraglutide, but Not Dapagliflozin or Acarbose, in Patients With Type 2 Diabetes: A 16-Week Randomized Parallel Comparative Study. Diabetes Care 2022, 45, 1201-1210, doi:10.2337/dc21-2064.

Reviewer 5 Report

I think this manuscript would be suitable for publication

Author Response

Thank you very much for your comments.